# Cell Type-Specific Anti-Adhesion Properties of Peritoneal Cell Treatment with Plasma-Activated Media (PAM)

**DOI:** 10.3390/biomedicines10040927

**Published:** 2022-04-18

**Authors:** Myriam Holl, Marie-Lena Rasch, Lucas Becker, Anna-Lena Keller, Laura Schultze-Rhonhof, Felix Ruoff, Markus Templin, Silke Keller, Felix Neis, Franziska Keßler, Jürgen Andress, Cornelia Bachmann, Bernhard Krämer, Katja Schenke-Layland, Sara Y. Brucker, Julia Marzi, Martin Weiss

**Affiliations:** 1Department of Women’s Health Tübingen, Eberhard Karls University Tübingen, 72076 Tübingen, Germany; myriam.holl@student.uni-tuebingen.de (M.H.); marie-lena.rasch@student.uni-tuebingen.de (M.-L.R.); laura.schultzerhonhof@nmi.de (L.S.-R.); felix.neis@med.uni-tuebingen.de (F.N.); franziska.kessler@med.uni-tuebingen.de (F.K.); juergen.andress@med.uni-tuebingen.de (J.A.); cornelia.bachmann@med.uni-tuebingen.de (C.B.); bernhard.kraemer@med.uni-tuebingen.de (B.K.); sara.brucker@med.uni-tuebingen.de (S.Y.B.); 2NMI Natural and Medical Sciences Institute, University Tübingen, 72770 Reutlingen, Germany; anna-lena.keller@nmi.de (A.-L.K.); felix.ruoff@nmi.de (F.R.); markus.templin@nmi.de (M.T.); silke.keller@uni-tuebingen.de (S.K.); katja.schenke-layland@nmi.de (K.S.-L.); julia.marzi@uni-tuebingen.de (J.M.); 3Institute of Biomedical Engineering, Eberhard Karls University Tübingen, 72076 Tübingen, Germany; lucas.becker@uni-tuebingen.de; 4Cluster of Excellence iFIT (EXC 2180) “Image-Guided and Functionally Instructed Tumor Therapies”, Eberhard Karls University, 72076 Tübingen, Germany; 5Department of Medicine/Cardiology, University of California, Los Angeles (UCLA), Los Angeles, CA 90095, USA

**Keywords:** physical plasma treatment, adhesion prophylaxis, postoperative adhesions, chronic inflammatory disease, cell type-specific response

## Abstract

Postoperative abdominal adhesions are responsible for serious clinical disorders. Administration of plasma-activated media (PAM) to cell type-specific modulated proliferation and protein biosynthesis is a promising therapeutic strategy to prevent pathological cell responses in the context of wound healing disorders. We analyzed PAM as a therapeutic option based on cell type-specific anti-adhesive responses. Primary human peritoneal fibroblasts and mesothelial cells were isolated, characterized and exposed to different PAM dosages. Cell type-specific PAM effects on different cell components were identified by contact- and marker-independent Raman imaging, followed by thorough validation by specific molecular biological methods. The investigation revealed cell type-specific molecular responses after PAM treatment, including significant cell growth retardation in peritoneal fibroblasts due to transient DNA damage, cell cycle arrest and apoptosis. We identified a therapeutic dose window wherein specifically pro-adhesive peritoneal fibroblasts were targeted, whereas peritoneal mesothelial cells retained their anti-adhesive potential of epithelial wound closure. Finally, we demonstrate that PAM treatment of peritoneal fibroblasts reduced the expression and secretion of pro-adhesive cytokines and extracellular matrix proteins. Altogether, we provide insights into biochemical PAM mechanisms which lead to cell type-specific pro-therapeutic cell responses. This may open the door for the prevention of pro-adhesive clinical disorders.

## 1. Introduction

The peritoneum is a thin mucosa lining the abdominal cavity and covers the intra-abdominal organs. It consists of a single layer of mesothelial cells and a loose stroma of connective tissue and fibroblasts, among others, directly beneath (Figure 1A) [1]. Despite numerous achievements in minimally invasive and open surgery, postoperative abdominal adhesions seriously limit the postoperative outcome and quality of life of patients and cause various serious clinical disorders due to the restriction of the mobility of affected abdominal organs. Peritoneal injury involves the recruitment, proliferation and activation of stromal cells such as fibroblasts and mesothelial cells. Inflammatory triggers such as cytokines lead to enhanced cell growth and increased secretion of pro-adhesive factors such as the ECM components collagen and fibronectin [1,2]. Having an incidence that ranges from 67% to 93%, abdominal adhesions are responsible for 15–20% of all cases of secondary infertility and for 50–70% of all mechanical ileus diseases, which are often characterized by severe clinical courses [3]. The annual costs of abdominal adhesions are estimated to be between USD 1.18 and 1.33 billion for the U.S. health system alone.

Current strategies for adhesion prevention are mainly based on barrier materials, which have not yet been able to show a clear clinical benefit [4]. Moreover, some of these are difficult to use, expensive and sometimes associated with serious side effects. Overall, there is a considerable need for effective technologies for adhesion prophylaxis to be routinely integrated into surgical procedures, as minimally invasive and open surgery are increasing in all medical disciplines worldwide [5]. A target of adhesion prophylaxis may be the reduction in cell proliferation as well as the synthesis and secretion of pro-adhesive factors.

Non-invasive physical plasma (NIPP) treatment is an emerging medical discipline. The treatment of liquid media with NIPP, which contains electrons, photons, ions as well as radical and nonradical reactive oxygen and nitrogen species (RONS) [6], results in the generation of plasma-activated medium (PAM). PAM maintains the major biological effects of NIPP, containing long-lived species, such as nitrite (NO_2−_), nitrate (NO_3−_) and hydrogen peroxide (H_2_O_2_) (Figure 1B). Previously, NIPP was shown to significantly improve wound healing and chronic inflammatory diseases as well as to induce promising cancer-selective anti-tumor effects in a broad variety of tumor tissues [7]. NIPP and PAM were able to influence proliferation and protein biosynthesis of connective tissue cells such as human fibroblasts in a dose-dependent manner [8].

In this study, we investigated the potential of PAM to modulate cell growth and pro-adhesive action of human peritoneal fibroblasts and mesothelial cells. We identified cell type-specific anti-proliferative cell responses by DNA interference, cell cycle arrest and apoptosis, accompanied with reduced synthesis and secretion of pro-adhesive proteins and cytokines. Our data indicate a high clinical potential of PAM to be used for comprehensive adhesion prophylaxis.

## 2. Materials and Methods

### 2.1. Cell Culture

Primary human fibroblasts and peritoneal wash cytology (PWC)-derived primary human mesothelial cells were isolated and cultured as previously described [9]. Ethical approval (Eberhard-Karls-University Tübingen): 649-2017BO2, approval: 12 January 2018; and 495/2018BO2, approval: 19 October 2018.

Immunofluorescence staining. Immunofluorescence (IF) staining on 1.4 × 10^4^ fibroblasts/cm^2^ and 2.9 × 10^4^ mesothelial cells/cm^2^ (µ-dish, 35 mm, 3.5 cm^2^ growth area, ibidi, Gräfelfing, Germany, #81158) was performed as previously described [9].

### 2.2. Preparation of Plasma-Activated Medium (PAM)

PAM was prepared by exposing plasma to 2 mL of Minimal Essential Medium (MEM) (Gibco™, Thermo Fisher Scientific Inc, Waltham, MA, USA, #31095029) without FCS and antibiotics. MEM was chosen to guarantee optimal cell growth conditions and to exclude other influences than those of plasma activation [9]. PAM exposure was performed in 6-well plates (9.6 cm^2^ growth area) using an ambient pressure argon plasma jet (kiNPen MED, neoplas med, Greifswald, Germany). Operating conditions: argon gas flow 4.0 L/min, frequency 1 MHz, line voltage 2–3 kV, power 1 W. The distance between the plasma source and the surface of medium was fixed at 7 mm using an external holding device and the duration for medium irradiation was set at 120 s to avoid significant influences of treatment distance and duration [8]. As characterized in previous studies, plasma treatment of cell culture fluids with kINPen MED is followed by a dose-dependent increase in RONS [8,10,11]. Then, 2 mL MEM were treated with pure argon gas and used as control treatment (ctrl). PAM was freshly produced for each experiment to limit degradation processes of short-living reactive plasma species.

### 2.3. Cell Confluency Assay

Primary isolated fibroblasts (6.25 × 10^3^/cm^2^) and mesothelial cells (4.7 × 10^4^/cm^2^) were cultivated for 24 h in a 96-well plate (0.32 cm^2^ growth area) and treated with 200 µL of indicated PAM dilutions for 4 h. After removing PAM, cells were washed with DPBS and cell confluence was observed by an IncuCyte S3-live cell imaging Systems (Essenbioscience, Göttingen, Germany) at 37 °C and 5% CO_2_ for 72 h. Confluency values were determined by the IncuCyte Software and normalized (relative confluence) to controls.

### 2.4. Raman Imaging

After PAM treatment, 2 × 10^4^ fibroblasts and 1.5 × 10^5^ mesothelial cells in imaging dishes (µ-dish, glass bottom, 3.5 cm^2^ growth area, ibidi, #81158) were fixed with 4% PFA for 10 min at 37 °C. After washing gently, cells were covered with DPBS. Raman imaging using a customized Raman microscope (alpha 300 R, WiTec GmbH, Ulm, Germany) equipped with a green laser (532 nm, maximum output power 60 mW) was performed as previously described [9].

### 2.5. Principal Component Analysis (PCA)

For in-depth analysis of molecular changes in nuclei, protein and lipid composition, high-intensity pixel representing nuclei, protein and lipid spectra were extracted from the Raman maps and applied for multivariate analysis as previously described [9].

### 2.6. Viability Assay

Primary isolated fibroblasts (2 × 10^3^ cells per well) and mesothelial cells (1.5 × 10^4^ cells per well) were seeded and cultivated for 24 h in a 96-well plate (0.32 cm^2^ growth area). Cells were treated with 200 µL of PAM for 4 h. The RealTime-Glo Cell Viability Assay (Promega, Fitchburg, WI, USA, # G9711) was performed according to the manufacturer’s instructions. The resulting luminescence signal was detected by a microplate reader (Spark, Tecan Trading AG, Männedorf, Switzerland). Values were normalized to controls.

### 2.7. Flow Cytometry

Flow cytometry was performed as previously described [9]. Applied specific antibodies: DSB-specific γH2AX formation: incubation of anti-γH2AX antibody, Ser139, JBW301 (Sigma-Aldrich, St. Louis, MO, USA) 1:125 dilution, 30 min on ice. Cell cycle phase analysis: incubation of DAPI (Sigma-Aldrich), 1:2 dilution, 30 min on ice. Forward and side scatter (FSC-H and SSC-H) characteristics were used to exclude debris. Forward scatter area and height (FSC-A and FSC-H) characteristics were used to exclude cell doublets (Appendix A).

### 2.8. Protein Expression Analysis by DigiWest Multiplex Protein Profiling

Analysis was performed by DigiWest multiplex protein profiling, as described previously [12]. The following primary antibodies were used: pH 3-specific antibody: protein kinase B phosphorylation (pAKT-specific antibody: 12,178 (D5G4), Cell Signaling Technology Cambridge, UK, 1:200), protein kinase B (AKT-specific antibody: 9272S, Cell Signaling Technology, 1:200), heat shock protein 27 (HSP27-specific antibody: 2402 (G31), Cell Signaling Technology, 1:200), Survivin (Survivin-specific antibody: 2808S (71G4B7), Cell Signaling Technology, 1:200), Signal transducer and activator of transcription 3 (STAT3-specific antibody: 9139S (124H6), Cell Signaling Technology, 1:200), Cyclin-dependent kinase 4 (CDK4-specific antibody: 12790S (D9G3E) (1272), Cyclin D1 (Cyclin D1-specific antibody: ab134175 (EPR2241), Abcam, Cambridge, UK, 1:200), cyclin-dependent kinase inhibitor 1 (p21-specific antibody: ab109520 (EPR362) Abcam, 1:200), p-Histone H3 (pH3-specific antibody: 9701, Cell Signaling Technology, 1:200), p53 phosphorylation (p-p53-specific antibody: 9284, Cell Signaling Technology, 1:200), Retinoblastoma protein (Rb-specific antibody: ab181616, Abcam, 1:200). Values of PAM-treated cells were normalized to the control group.

### 2.9. Apoptosis; Caspase 3/7 Assay

Primary isolated fibroblasts (2 × 10^3^ cells per well) and mesothelial cells (1.5 × 10^4^ cells per well) were seeded and cultivated for 24 h in a 96-well plate (0.32 cm^2^ growth area). The Caspase 3/7 assay (Essen Bioscience, Sartorius, Göttingen, Germany, #4704) was performed according to the manufacturer’s instructions. The generated fluorescence signals were detected by IncuCyte S3-live cell imaging Systems after 24 and 72 h. Results were normalized to controls.

### 2.10. DNA Methylation; 5mC Staining

For analyzation of the global genomic 5mC methylation status, IF staining was performed by use of a 5mC-specific mouse monoclonal IgG antibody (MABE146, diluted in 0.1% BSA in PBS at 1:2000 ratio, Merck, Darmstadt, Germany) as previously described [13]. Then, 2 × 10^4^ PAM-treated fibroblasts were incubated in 6-well glass bottom cell culture plates (ibidi, 6 mL DMEM/well) for the respective time periods.

### 2.11. Western Blot

Fibroblasts were seeded in 100 mm cell culture dishes (56.7 cm^2^ growth area) with 3.4 × 10^5^ cells/dish. Then, 120 h after treatment, supernatant was discarded and cells were frozen at −80 °C. Ice-cold TRIS-HCl cell lysis buffer was added, cells were harvested using a cell scraper. The lysate was incubated on ice for 30 min before centrifugation for 15 min at 13,000 rpm at 4 °C and BCA-Protein Assay (Thermo Fisher Scientific, Waltham, MA, USA, #23227) using a microplate reader (Spark, Tecan Trading AG, Switzerland) following the manufacturer’s instructions. Samples were denaturized in 4× Lämmli protein sample buffer (BioRad, Hercules, CA, USA, #1610747), diluted with 10% beta-mercaptoethanol, for 10 min at 95 °C. Gel electrophoresis was performed in XCell SureLock Mini-Cell Electrophoresis Chambers using NOVEX NuPage 4–12% Bis-Tris Protein Gels, 1.0 mm and MES Running Buffer (20×) in ddH_2_O (all Thermo Scientific Inc.). For protein blotting, nitrocellulose membranes were soaked in NuPAGEtrade Transfer Buffer (Thermo Scientific Inc., #NP00061). Blotting was performed using a SemiDry Transfer System (peqLab, VWR International, Radnor, PA, USA) for 2 h at constant 0.2 A. After washing, membranes were incubated overnight at 4 °C with primary antibodies under constant agitation: fibronectin (1:1000 in 0.1% BSA, abcam), collagen I-alpha antibody (1:1000 in 0.1% BSA, NovusBiologicals, Littleton, CO, USA) and GAPDH (1:1000 in 0.1% BSA, Cell Signaling 14C10). For collagen I alpha Western blot, membranes were thawed and then blocked with 5% BSA in 1× PBS for 45 min. Membranes were incubated for 2 h at room temperature with the secondary antibody AlexaFluor488 (1:10,000 in 0.1% BSA) in the dark. For detection, the Amersham™ Typhoon™ Biomolecular Imager (Cytiva, Marlborough, MA, USA) was used. Membranes were frozen at −20 °C until further analysis.

### 2.12. Hydroxyproline Assay

The insoluble collagen content of the ECM was determined according to Keller et al. and Capella-Monsonís et al. [14,15] by using the Hydroxyprolin-Assay by Sigma-Aldrich (Merck Milipore, Burlington, MA, USA). All samples and standards were run in duplicates. Cells were seeded into 100 mm cell culture dishes (56.7 cm^2^ growth area) with 2.4 × 10^4^ cells/cm^2^. The analysis of ECM components was performed 120 h after treatment in order to synthesize sufficient components for detection.

### 2.13. Soluble Collagen Assay

For the assessment of the total soluble collagen content of the ECM, the Sircol™ Soluble Collagen Assay kit (Biocolor, Carrickfergus, County Antrim, UK, S2000) was used according to the manufacturer’s information. Cells were seeded in 75 cm^2^ cell culture flasks with 6 × 10^3^ cells/cm^2^. The analysis of ECM components was performed 120 h after treatment in order to synthesize sufficient components for detection. Samples’ absorbance for soluble collagen content was measured at 555 nm using a Spark microplate reader (Tecan, Männedorf, Swizzerland).

### 2.14. Matrix Metalloproteinases (MMPs) Assay

Fibroblasts were seeded in 6-well plates (9.6 cm^2^ growth area) with a density of 2.5 × 10^5^ cells per well. After PAM treatment, the medium was frozen at −80 °C. Human ELISA kits for MMP-1 and MMP-2 were both obtained from Thermo Fisher Scientific. The manufacturer’s protocol was followed, and samples were measured at 450 nm using a Spark microplate reader (Tecan).

### 2.15. Cytokine Multiplex Assay

Cytokines were analyzed using the Multiplex system. Fibroblasts were seeded in 24-well plates (1.9 cm^2^ growth area) with a density of 5 × 10^4^ cells per well. A standardized scratch before PAM treatment served as imitated intraoperative injury. Supernatant was collected and frozen immediately at −80 °C before analysis.

### 2.16. Statistical Analysis

Statistical comparison was carried out with Student’s *t*-test or ANOVA (GraphPad Prism version 9.0, GraphPad Software, San Diego, CA, USA), as specified in the figure legends. The data are expressed as mean ± standard deviation. *p*-values < 0.05 were considered statistically significant. Experiments were performed in at least three independent experimental approaches.

## 3. Results

### 3.1. Cell Type-Specific Anti-Proliferation

Postoperative intra-abdominal adhesions severely influence daily life and surgical outcome by interfering with the mobility of the abdominal organs. Peritoneal mesothelial cells and fibroblasts play important and complex roles in wound healing and adhesiogenesis. RONS were reported to be the main effective compounds of PAM which are formed by plasma irradiation at the interface between plasma discharge and the surrounding gas and liquid phases (Figure 1B). In order to analyze cell-specific anti-adhesive PAM effects, we used human peritoneal mesothelial cells and fibroblasts isolated from peritoneal wash cytologies and peritoneal tissue samples as previously described by our group (Figure 1C,D) [9]. Using this patient-specific 2D in vitro model, we demonstrated cell type-specific and dose-defined anti-proliferative efficacy of PAM. PAM-treated fibroblasts exhibited a significantly enhanced decrease in relative cell confluency compared to mesothelial cells (Figure 1E,F). The PAM dilution 1:2 was able to selectively target the pro-adhesive fibroblasts, whereas the proliferation of mesothelial cells was preserved (Figure 1G,H). In conclusion, we assumed the feasibility to selectively inhibit the pro-adhesive cell pattern of fibroblasts while maintaining the wound closure ability of mesothelial cells.

### 3.2. PAM Treatment Induces Molecular Alterations of Essential Cell Components while Maintaining Cell Morphology

Contact- and marker-free Raman imaging was applied as a non-destructive molecular fingerprint analysis enabling the identification, localization and biochemical assessment of molecular PAM impact on different cell components (Figure 2A) [16,17]. Using true component analysis (TCA) based on specific Raman signatures, nuclei (blue), cytoplasmic proteins (green) and lipids (yellow) were localized by generating false color-coded intensity distribution heat maps for each cellular component (Figure 2B,C). No distinct effect on the morphological integrity of peritoneal cells could be observed after PAM treatment (Figure 2C). In-depth molecular multivariate analysis of nuclear, cytoplasmic and lipid features demonstrated a clear separation between fibroblasts and mesothelial cells and the controls (PC-1 vs. PC-2 scores) (Figure 2D,E and Appendix A). Loading plots of the underlying spectral information allowed the assignment to corresponding molecular groups based on shifts in the spectral signature (Figure 2F,G and Appendix A). Relevant peaks are summarized in Appendix A [18,19,20,21,22,23,24,25,26,27,28,29,30,31,32,33,34,35,36,37,38,39,40]. The molecular changes in PAM-treated fibroblasts and mesothelial cells were especially attributed to peak shifts of DNA bases, cell membrane lipids and cytoplasmatic proteins, some of which were previously reported in the context of methylation and apoptosis [41,42]. Score plot analysis of these components revealed statistically significant differences between PAM-treated cells and controls (Figure 2H,I). Taken together, the Raman data indicate that PAM leads to molecular changes, especially in DNA structure, cell membrane and protein expression, independent of the cell type. This molecular fingerprint from Raman imaging was further analyzed in depth by specific molecular biological methods.

### 3.3. PAM Induces Cell Type-Specific Anti-Proliferative Signaling

We next investigated the molecular cell response of mesothelial cells and fibroblasts in terms of cell viability, induction of DNA damage as well as cell cycle and apoptosis regulation. For this, both peritoneal cell types were treated with the cell type-selective PAM dosage 1:2. By measuring intracellular adenosine triphosphate (ATP), we found that PAM treatment significantly decreased cell viability in fibroblasts while that of mesothelial cells was not affected (Figure 3A). Because of the reactive properties of RONS which frequently interact with nucleic acids, the induction of complex cellular recognition and repair mechanisms for DNA damage has been frequently described after direct plasma treatment [43]. To semi-quantify DNA damage following PAM treatment, we used cytometry analysis after cellular staining with specific antibodies against phosphorylated histone H2AX at Ser139, which is a commonly used biomarker for DNA double-strand breaks. Only in fibroblasts, PAM treatment was followed by a statistically significant rapid and persistent H2AX phosphorylation within 72 h (Figure 3B). Cell cycle analysis by propidium iodide staining and cytometry within 72 h revealed a significant increase in fibroblasts in the G2 phase (Figure 3C). Again, this effect could not be demonstrated in mesothelial cells (Figure 3D). DigiWest protein profiling of fibroblasts revealed a significant increase in cell cycle-regulating proteins such as cyclin-dependent kinase 4 (CDK4) and cyclin D1 as well as cyclin-dependent kinase inhibitor 1 (CDKN1A, p21) [44], accompanied by a crucial decrease in the mitotic biomarker phospho-histone H3 (Ser10) [45] (Figure 3E,F). As a result, we reasoned that a substantial subset of the fibroblast cell population was arrested in the G2 phase and did not enter mitosis [45]. Given that PAM induces DNA damage, followed by cell cycle arrest, we assumed that apoptotic pathways may occur as a physiological consequence and a known PAM effect [46]. Indeed, and only in fibroblasts, PAM treatment resulted in activation of the effector cysteine-dependent aspartate-specific proteases (caspases) casp3 and casp7, p53 phosphorylation and activation (pp53) as well as the suppression of the tumor suppressor retinoblastoma protein (Rb) (Figure 3G,H). In contrast, pivotal anti-apoptotic factors such as protein kinase B (AKT) and phosphorylated AKT (pAKT), heat shock protein 27 (HSP27) and BIRC5 (survivin) were induced in mesothelial cells (Figure 3H).

### 3.4. PAM Treatment Reduces Pro-Adhesive Protein Expression

We next sought to characterize the impact of PAM on gene and protein expression, with a focus on pro-adhesive factors critical for adhesion development. To determine the global level of gene expression after PAM treatment, we examined the overall DNA methylation status by immunofluorescence (IF) staining with specific antibodies against 5-methylcytosine (5mC). DNA methylation regulates gene expression and typically represses gene transcription. PAM treatment was accompanied by significantly increased gene methylation patterns (Figure 4A,B). This was supported by the significant increase in the DNA methyltransferase 1 (DNMT1) activity (Figure 4C). We next evaluated the expression of typical pro-adhesive factors and cytokines by DigiWest protein profiling and a cytokine multiplex assay. Reductions in both fibroblast growth factor 10 (FGF-10) and FGF receptor were demonstrated (Figure 4D). Furthermore, we found a suppression of collagen I and fibronectin expression (Figure 4D) as well as a reduced secretion of the pro-inflammatory cytokines granulocyte-macrophage colony-stimulating factor (GM-CSF) and interleukin 1 beta (IL-1b) (Figure 4E). The former was confirmed by confocal IF microscopy and fibronectin-specific semiquantitative Western blot and chemiluminescence analysis (Figure 4F–K and Appendix A). It is known that 90% of the collagen present in the body is of type I [47]. A significant portion of amino acids in collagen I represents hydroxyproline [48]. We analyzed the synthesized ECM after 120 h (Figure 4L) according to Keller et al. [15,49] and found that hydroxyproline was significantly reduced after single PAM treatment. Indirectly, this also suggests that the concentration of extracellular collagen I is significantly reduced after PAM treatment. Matrix metalloproteinases (MMPs) are proteolytic enzymes with a crucial role in the ECM remodeling process during tissue fibrosis and the development of postoperative adhesions [3]. Fibroblasts demonstrated a statistically significant decrease in MMP-2 and a tendential decrease in MMP-1 expression within 72 h after PAM treatment measured by ELISA (Figure 4M and Appendix A). PAM-treated mesothelial cells revealed a significant downregulation of MMP-1 after 72 h (Appendix A). Furthermore, the strong and stable downregulation of MMPs in fibroblasts was accompanied by a reduction in transforming growth factor beta 1 (TGF-β1) expression (Figure 4N).

Taken together, these data indicate that PAM treatment significantly reduces the expression and secretion of relevant adhesive factors and interferes with multiple intracellular regulatory mechanisms that, if untreated, critically contribute to the pro-adhesive capacity of fibroblasts (Figure 5).

## 4. Discussion

In the present study, PAM treatment resulted in efficient anti-proliferative effects at the molecular and cellular levels, which were cell type-specific depending on the PAM dosage. The investigation of selective plasma effects has been the subject of several in vitro studies, based on which a few mechanistic models have been established regarding cancer cells.

In this study, we transferred the well-known PAM effects on neoplastic cells to primary human peritoneal tissue, the emergence site of postoperative intra-abdominal adhesions. Fibroblasts and mesothelial cells were isolated from solid peritoneal tissue and peritoneal wash cytology, respectively [9]. In this setting, MEM was used for plasma treatment to provide optimal conditions for cell growth and metabolism. Future studies need to confirm the study results by using medically approved and intracorporeally applicable substances and buffers.

By Raman imaging, we could exclude any changes in cell morphology after PAM treatment. The significant molecular changes, however, were predominantly assigned to peak shifts of DNA bases, cytosolic proteins and cell membrane lipids. Using standard molecular biology methods, we confirmed selective PAM effects on fibroblasts’ proliferation and metabolism. In this regard, DNA damage was accompanied by cell cycle arrest and apoptosis. We demonstrated both very early DNA damage events (most probably through RONS-driven chemical modifications of DNA molecules) and late DNA damage events after 72 h (which could be a consequence of apoptotic processes). Following ROS formation and DNA double-strand breaks we found alterations in genomic methylation patterns and p53-associated signal transduction cascades. In line with current literature, we found that a subset of the cell population remains in the G2 phase [50], supported by cell cycle-specific molecular markers (downregulation of mitosis-specific histone H3 phosphorylation; increase in associated cell cycle regulators CDK4, cyclin D1 and p21) [44,45]. As a consequence, PAM-treated fibroblasts entered apoptosis, shown by the tendential activation of the effector caspases casp3/7. Mesothelial cells initiated the intrinsic survival program (AKT-pathway, HSP27, survivin).

Fibroblasts involved in peritoneal adhesions show increased proliferation and protein titration as well as reduced apoptosis rates [51,52,53]. Moreover, increased basal mRNA and protein levels of collagen I, fibronectin, MMPs and TGF-β 1/2 interleukins were reported [54]. TGF-β1 is a potent trigger of epithelial mesenchymal transition (EMT) that promotes the loss of epithelial features, including apico-basal polarity, intercellular contacts and the gain of mesenchymal features, including increased migratory capacity and contractility [55]. Beyond this, it was shown that TGF-β induces collagen I overexpression followed by skin keloids [56]. In this study, PAM was able to decrease extracellular levels of TGF-β1 and IL-1. The blockade of TGF-β has been a successful approach to prevent adhesions in animal experiments [57,58,59]. In addition, the intracellular expression of pro-adhesive fibronectin and partially that of collagen I was reduced after PAM treatment. In accordance, Sung et al. recently reported a reduced expression of collagen I after plasma treatment of fibroblasts derived from skin keloids [60]. Collagen I is a main part in ECM and an important factor in peritoneal adhesions. The inhibition of collagen I deposition and fibroblast proliferation with dexamethasone was the subject of previous studies to effectively prevent postoperative adhesions in the mouse model [61]. Collagen I proteins are composed of a triple helix [62]. Characteristic of collagen is its high content of hydroxyproline amino acids, which has made hydroxyproline a common measure of total collagen I and collagen I metabolism [48,63]. Clinical studies applied collagen density measurements, hydroxyproline content determination, fibroblast counts and densitometric analysis of collagen I to evaluate the treatment efficacy in reducing intra-articular adhesion [64]. The reduction in hydroxyproline content thereby strongly correlated with the reduction in postinterventional adhesions. The ability of PAM to significantly decrease the extracellular hydroxyproline concentration suggests that PAM (i) significantly reduces extracellular collagen I and total ECM and, therefore, (ii) could positively correlate with reduced formation of adhesions in vivo.

In future, PAM may be applied intraoperatively during or immediately following the surgical activities using a spray device. In this early phase, fibroblast-rich stroma is exposed and induces pro-adhesiogenic processes. Here, PAM may effectively intervene by selectively inhibiting fibroblasts having contact with the intra-abdominal space. Deeper tissue layers as well as surrounding mesothelial cells remain unaffected and may enable enhanced peritoneal wound closure. The pathogenesis of adhesion formation is complex and is not limited to the cellular component. Additional mechanisms involved are systemic coagulopathy and fibrin deposits, which form a matrix for the development of fibro-collagenous tissue. Besides the present work focusing on cellular plasma responses, previous studies showed that direct plasma treatment has a significant impact on non-cellular components, such as blood hemostasis resulting in shortened full clotting time [65]. Moreover, the present study is limited to 2D cell culture experiments. Previous studies found that the “activation status” of fibroblasts is not completely reliable in these systems. The contact with surface polymers of cell culture materials may be followed by activation of fibroblasts [66]. Due to this, the 2D cell culture in vitro data must be validated under in vivo (-like) conditions.

## 5. Conclusions

Here, we report the first application of PAM on human peritoneal tissue to prevent pro-adhesive cell responses. Based on the results of the present study, the clinical application of PAM may represent a promising method to limit or prevent the formation of postoperative adhesions. Peri- or postoperative flushing of the abdominal cavity with PAM could suppress the dysregulation of pro-adhesive ECM-producing connective tissue cells.

## Figures and Tables

**Figure 1 biomedicines-10-00927-f001:**
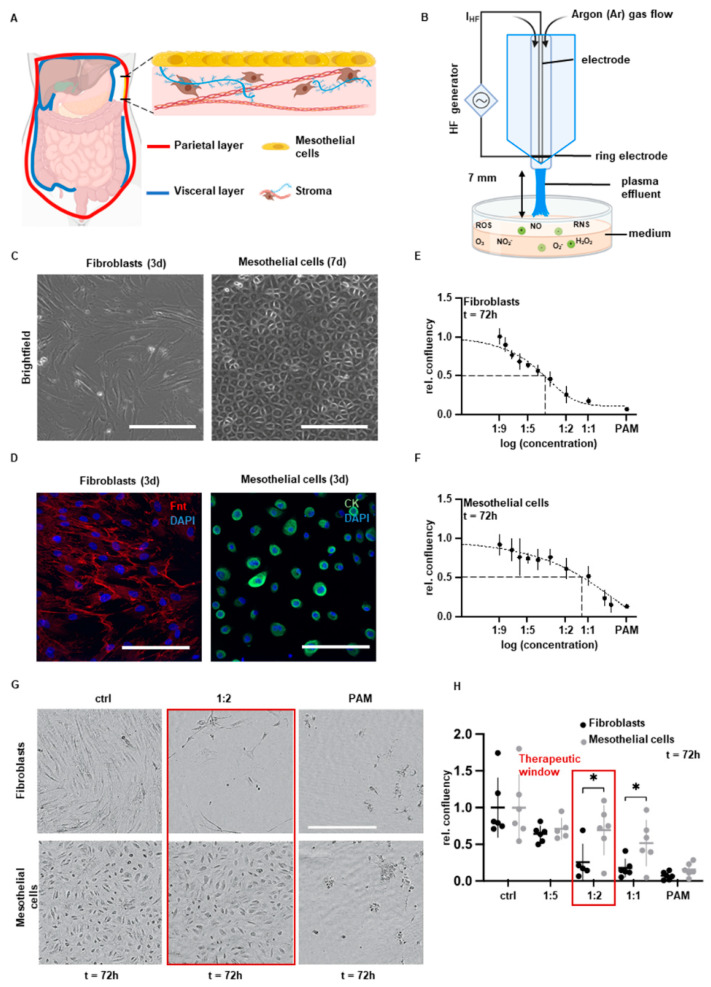
Characterization of cell type-specific growth inhibition in primary human peritoneal cells. (**A**) Schematic of anatomical and histological features of the peritoneum. (**B**) Schematic of the experimental setup of PAM generation. (**C**) Representative brightfield microscopy of native primary fibroblasts and mesothelial cells. Scale bar represents 200 µm. (**D**) Representative IF microscopy of fibroblasts and mesothelial cells after PFA fixation and staining with specific antibodies against cytokeratin and fibronectin. Scale bar represents 200 µm. (**E**,**F**) Relative cell confluency of (**E**) fibroblasts and (**F**) mesothelial cells 72 h after incubation of different PAM dosages for 4 h (mean ± SD). (**G**) Representative brightfield microscopy of fibroblasts and mesothelial cells 72 h after 4 h incubation with indicated PAM dilutions for 4 h and control treatment. Scale bar represents 400 µm. (**H**) Relative cell confluency 72 h after incubation with indicated PAM dilutions for 4 h with and control treatment (mean ± SD; * *p* < 0.05; paired *t*-test).

**Figure 2 biomedicines-10-00927-f002:**
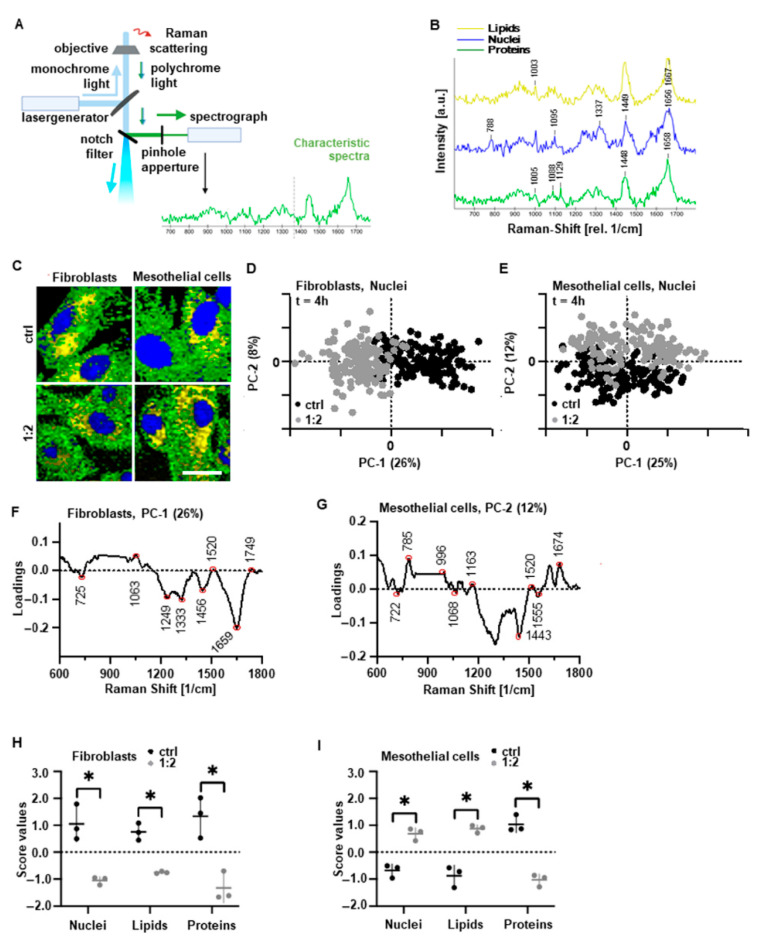
Characterization of cellular PAM effects using contact- and label-independent Raman imaging. Cells were incubated with 1:2 diluted PAM for 4 h and analyzed after indicated timepoints. (**A**) Schematic of the Raman microscope and an exemplary Raman spectrum representing a specific biomolecule. (**B**) Representative Raman spectra and characteristic bands that were used to identify the molecular components. (**C**) Raman intensity distribution heat maps assigned to nuclei (blue), lipids (yellow) and cytoplasmic proteins (green) of fibroblasts and mesothelial cells after 4 h of PAM incubation; scale bar represents 50 µm. (**D**,**E**) PCA demonstrated a separation in the PC-1 vs. PC-2 score plot for the nuclei component in fibroblasts (**D**) and mesothelial cells (**E**). (**F**,**G**) Corresponding PC-1 and PC-2 loading plot for the nuclei component indicating changes in DNA after PAM treatment of (**F**) fibroblasts and (**G**) mesothelial cells. (**H**,**I**) Statistical comparison (of the spectra for nuclei, lipids and cytoplasmatic proteins obtained in (**C**)) was performed by PCA and subsequent normalization of the PC score values to the mean values of controls. This enabled the assessment of molecular differences in nuclear, lipid and cytosolic protein composition; the data points represent average score values per donor (mean ± SD; * *p* < 0.05; paired *t*-test).

**Figure 3 biomedicines-10-00927-f003:**
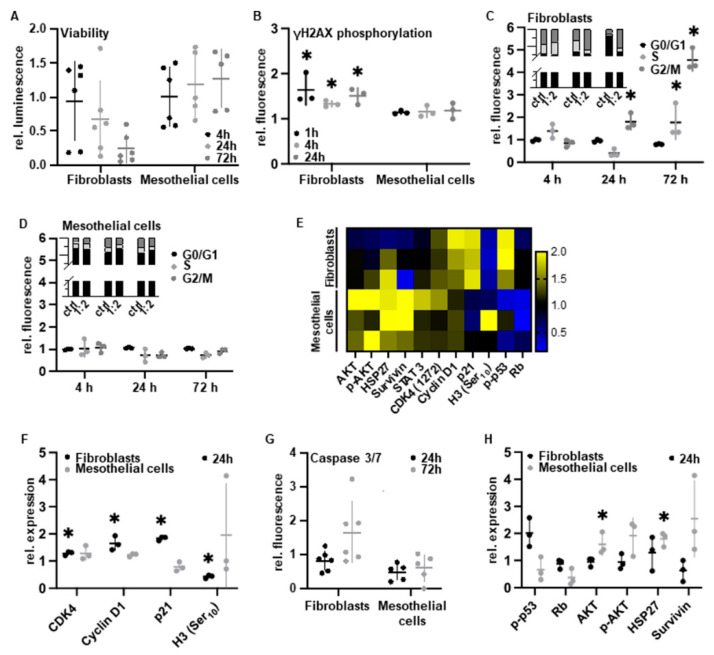
PAM treatment induces anti-proliferative pathways in fibroblasts and initiates cell survival in mesothelial cells. Cells were incubated with 1:2 diluted PAM for 4 h and analyzed after indicated timepoints. (**A**) Relative cell viability after 4 h of PAM incubation relative to controls. (**B**) Relative γH2AX intensity in flow cytometry after PAM incubation relative to controls. (**C**,**D**) Relative flow cytometry fractions of cells in cell cycle phases S, G0/G1 and G2/M after PAM incubation in (**C**) fibroblasts and (**D**) mesothelial cells relative to controls. (**E**,**F**,**H**) DigiWest protein profiles of fibroblasts and mesothelial cells after PAM incubation relative to controls. (**E**) Heat map of log2 transformed DigiWest data. Data were median-centered, and hierarchical clustering was performed using complete linkage and Euclidean distance, utilizing the MultiExperiment Viewer (MeV version 4.9.0, [45]) software. Yellow indicates a high signal level; blue indicates a low signal level (compared to median). (**F**) Relative expression of the cell cycle-regulating factors CDK4, Cyclin D1, p21 and H3 (Ser10) in fibroblasts (black) and mesothelial cells (gray). (**G**) Relative caspase-3/7 activity after PAM incubation relative to controls. (**H**) Relative expression of the anti-proliferative and pro-apoptotic factors p-p53 and Rb and the cell survival factors AKT, p AKT, HSP27 and survivin in fibroblasts (black) and mesothelial cells (gray). Results are expressed as mean ± SD; * *p* < 0.05 as determined by paired *t*-test.

**Figure 4 biomedicines-10-00927-f004:**
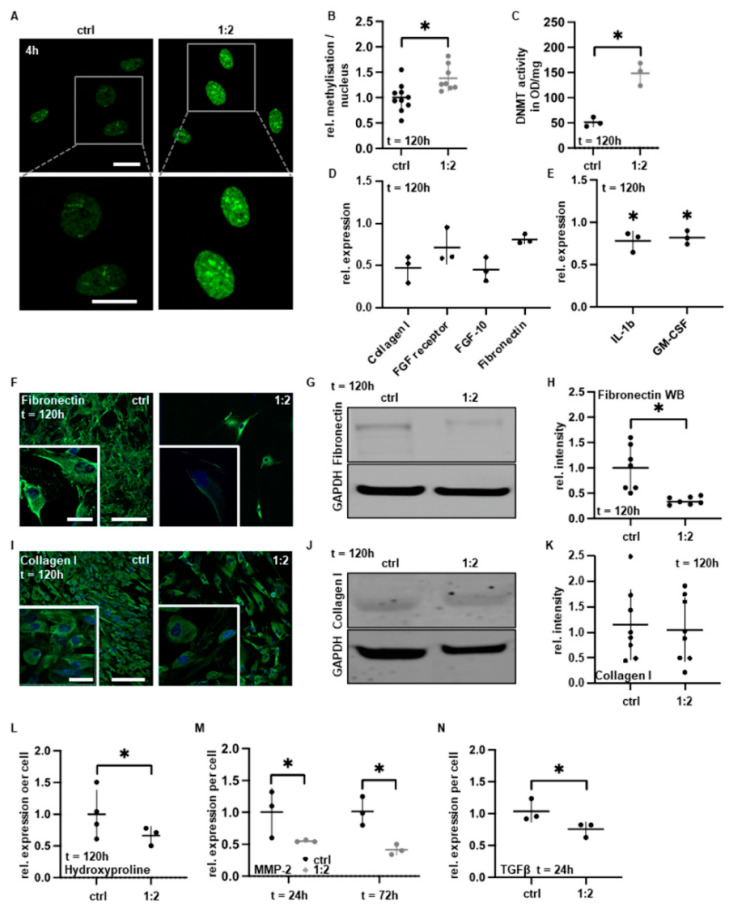
PAM treatment attenuates protein biosynthesis and the secretion of pro-adhesive factors. Fibroblasts were incubated with 1:2 diluted PAM for 4 h and analyzed after indicated timepoints. (**A**) Representative IF microscopy after staining with anti-5mC antibodies and (**B**) relative genomic methylation level per nucleus (number of foci normalized to the control); the scale bar equals 10 µm. (**C**) DNMT activity level per cell. (**D**) DigiWest protein profiles of pro-adhesive factors collagen I, FGF receptor and FGF-10 and fibronectin relative to controls. (**E**) Cytokine multiplex assay of GM-CSF and IL-1b in fibroblast supernatants relative to control. (**F**,**G**) Representative IF microscopy of fibronectin (**F**). Scale bars represent 100 and 10 µm, respectively. (**G**,**H**) Representative Western blot of fibronectin (**G**), and relative of fibronectin expression (**H**) (analyzed from (**G**)). (**I**,**J**) Representative IF microscopy of collagen I (**I**). Scale bars represent 100 and 10 µm, respectively. (**J**,**K**) Representative Western blot of collagen I (**J**), and relative collagen I expression (**K**) (analyzed from (**J**)). (**L**) Relative extracellular hydroxyproline expression. (**M**) Relative MMP-2 expression. (**N**) Relative expression of TGF β. Results are expressed as mean ± SD; * *p* < 0.05 as determined by paired *t*-test.

**Figure 5 biomedicines-10-00927-f005:**
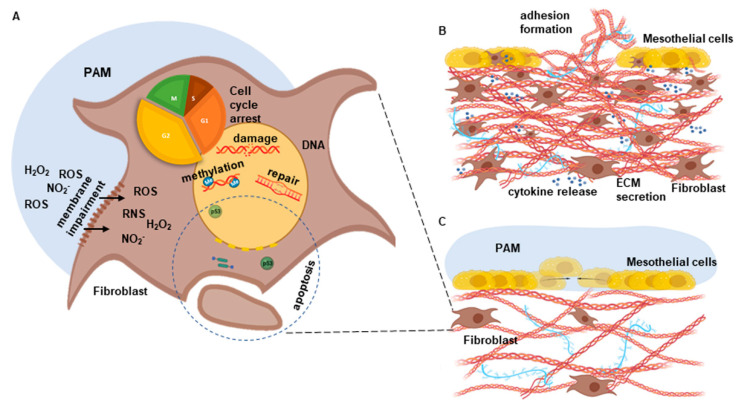
(**A**) Reactive species in PAM induce an intracellular increase in ROS and RNS, especially by cytoplasmic membrane impairment. This is followed by the induction of various intracellular response pathways such as altered genomic methylation patterns and signal transduction cascades involving attenuation of cell growth and protein biosynthesis by cell cycle arrest and p53-associated apoptosis. (**B**) Schematic model of pathological cell growth, cytokine secretion and secretion of pro-adhesive molecules such as collagen and fibronectin following peritoneal disruption of the superficial cell layer. (**C**) Hypothesized mode of action of PAM application including inhibition of fibroblast proliferation and attenuation of cytokine and ECM components secretion, and unhindered re-epithelialization by mesothelial cells.

## Data Availability

The data presented in this study are available on request from the corresponding author. The data are not publicly available due to the ongoing character of the research project.

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
