# Peer review of "Cell Type-Specific Anti-Adhesion Properties of Peritoneal Cell Treatment with Plasma-Activated Media (PAM)"

_biomedicines, 2022, doi:10.3390/biomedicines10040927_

Round 1

Reviewer 1 Report

Dear Authors,

The manuscript biomedicines-1653462, entitle ‘Cell type-specific anti-adhesion properties of peritoneal cell treatment with plasma-activated media (PAM)’ presents experimental results on the effects of anti-adherence of peritoneal cells after PAM (plasma activated medium) exposure.

The authors try to modulate the proliferation and protein biosynthesis of these cell by PAM exposure. Their protocol includes the usage of several methods of investigation, such as: immunofluorescence staining, cell confluency assay, Raman imaging, viability assay, flow-cytometry, protein expression analysis via protein profiling, apoptosis - Caspase 3/7 assay, DNA methylation - 5mC staining, western blot, hydroxyproline assay, soluble collagen assay, matrix metalloproteinases (MMPs) assay, cytokine multiplex assay, as well as statistical analysis (e.g. PCA, Student’s t-test or ANOVA). These mentioned techniques are well described and their results well corelated.

The main manuscript body is well written and well organized. The conclusions are well defined and supported by the experimental findings.

I do, however, have some suggestions that will make the readability of this manuscript better to the untrained reader (unspecialized):

  1. The PAM production is scarce presented on page 2, between lines 84-92. Maybe a separate paragraph with more details on how the medium was chosen, the treatment distance / duration, why ‘fresh’ PAM, why not ‘old’ – preserved PAM? References should be added here on these conditions. Are there significant influences of treatment distance and duration upon PAM production? How the chosen medium is activated? What does it change: pH, RONS, conductivity? These should be added to this ‘PAM production’ paragraph, along with references.
  2. On page 6, figure 1: I would suggest that this figure be bigger, like full page width as it contains many detailed that must be identified easily by the reader;

After this info added to the manuscript, I consider that it could be taken into consideration to be published in Biomedicines Journal.

Accept with MINOR REVISION

Author Response

We want to thank the reviewers for their constructive comments and suggestions, which clearly helped to further improve the work. Please find our detailed point-by-point responses below. We carefully adjusted our manuscript according to the reviewers’ comments. 

Response to the reviewers

Reviewer # 1, Comment 1: The PAM production is scarce presented on page 2, between lines 84-92. Maybe a separate paragraph with more details on how the medium was chosen, the treatment distance / duration, why ‘fresh’ PAM, why not ‘old’ – preserved PAM? References should be added here on these conditions. Are there significant influences of treatment distance and duration upon PAM production? How the chosen medium is activated? What does it change: pH, RONS, conductivity? These should be added to this ‘PAM production’ paragraph, along with references.

Authors’ response: We thank reviewer #1 for his positive statement and suggestions. We agree that a separate paragraph with more details on the mentioned questions is beneficial. Thus, we have added a separate paragraph “2.2. Preparation of plasma-activated medium (PAM)”, addressing the issues suggested and citing concerning literature:

“2.2. Preparation of plasma-activated medium (PAM)

PAM was prepared by exposing plasma to 2 mL of Minimal Essential Medium (MEM) (Gibco™, Thermo Fisher Scientific Inc, Waltham, MA, USA, #31095029) without FCS and antibiotics. MEM was chosen to guarantee optimal cell growth conditions and to exclude other influences than those of plasma activation [9]. PAM exposure was performed in 6-well plates (9.6 cm2 growth area) using an ambient pressure argon plasma jet (kiNPen MED, neoplas med, Greifswald, Germany). Operating conditions: argon gas flow 4,0 L/min, frequency 1 MHz, line voltage 2-3 kV, power 1 W. The distance between the plasma source and the surface of medium was fixed at 7 mm using an external holding device and the duration for medium irradiation was set at 120 s to avoid significant influences of treatment distance and duration [8]. As characterized in previous studies plasma treatment of cell culture fluids with kINPen MED is followed by a dose-dependent increase in RONS [8,11]. 2 mL MEM were treated with pure argon gas and used as control treatment (ctrl). PAM was freshly produced for each experiment to limit degradation processes of short-living reactive plasma species.”

Reviewer #1, Comment 2: On page 6, figure 1: I would suggest that this figure be bigger, like full page width as it contains many detailed that must be identified easily by the reader.

Authors’ response, Comment 2: We agree with this comment and enlarged Figure 1.

Reviewer 2 Report

Postoperative adhesions remain one of the more challenging issues in surgical practice. Although the majority of postoperative adhesions are clinically silent, the consequences of adhesion formation can represent a lifelong problem including chronic abdominal pain, recurrent intestinal obstruction requiring multiple hospitalizations, and infertility. Despite recent advances in surgical techniques, there is no reliable strategy to manage postoperative adhesions. Therefore, the problem that the authors' work is aimed at is extremely relevant. The work was done at a high methodological level. However, the pathogenesis of adhesion formation is complex and far from being limited to the cellular component. The systemic coagulopathy results in the fibrin deposits, which are a matrix for development of fibrocollagenous tissue and formation of an extracellular matrix.

Accordingly, it is unclear how to use PAM to treat adhesions. At the moment, the introduction does not correlate with the work done. It is possible to supplement the work with an analysis of the effect of the applied plasma regimens on blood clotting, fibrogenesis, as an option.

Also, the process of adhesion formation begins in the early postoperative period. It is not entirely clear how the use of plasma can help, as they have been shown to inhibit proliferation and induce cell apoptosis. But during the postoperative period, one should not forget about healing, to what extent this can affect the process of regeneration of the operated organ. Therefore, I strongly advise you to reconsider the general concept of work.

Fibroblasts involved in peritoneal adhesions show increased proliferation and protein secration as well as reduced apoptosis rates.  It would be interesting to check how much plasma normalizes the excessive proliferation of fibroblasts activated, for example, by cytokines, and not calm state fibrablasts.

There are also some small remarks

  • «Forward- and side-scatter (FSC-H and SSC-H) characteristics were used to exclude dead cells» (Line 122-123). Dead cells can be eliminated using special dyes such as propidium iodide. In this case, it is debris - the remains of cells

Figure 1D it is not clear where the marker is. Please indicate either in the 

Author Response

We want to thank the reviewers for their constructive comments and suggestions, which clearly helped to further improve the work. Please find our detailed point-by-point responses below. We carefully adjusted our manuscript according to the reviewers’ comments.

Response to the reviewers

Reviewer #2, Comment 1: Accordingly, it is unclear how to use PAM to treat adhesions. At the moment, the introduction does not correlate with the work done. It is possible to supplement the work with an analysis of the effect of the applied plasma regimens on blood clotting, fibrogenesis, as an option.

Also, the process of adhesion formation begins in the early postoperative period. It is not entirely clear how the use of plasma can help, as they have been shown to inhibit proliferation and induce cell apoptosis. But during the postoperative period, one should not forget about healing, to what extent this can affect the process of regeneration of the operated organ. Therefore, I strongly advise you to reconsider the general concept of work.

Authors’ response, Comment 1: We thank Reviewer #2 for his comments. However, we regret his statement that the general concept of work should be reconsidered. We feel that this is a personal opinion decoupled from the actual data presented in this manuscript.

To avoid misunderstandings and to take the comments of Reviewer #2 into account, we included a passage into the discussion section, where we explain the future intraoperative application of PAM. Furthermore, we included information about which peritoneal tissue layers are involved and how these be affected from PAM treatment. Moreover, we draw attention to the correct remark of Reviewer #2 that the formation of adhesions is not based solely on cellular factors. Therefore, we refer to other work showing that plasma also exerts an influence on non-cellular components involved in adhesion formation. However, the present research work focusses on cellular plasma responses.

“In future, PAM may be applicated intraoperatively during or immediately following the surgical activities using a spray-device for example. In this early phase, fibroblast-rich stroma is exposed and induces pro-adhesiogenic processes. Here, PAM may effectively intervene by selectively inhibiting fibroblasts having contact with the intra-abdominal space. Deeper tissue layers as well as surrounding mesothelial cells remain unaffected and may enable enhanced peritoneal wound closure. The pathogenesis of adhesion formation is complex and is not limited to the cellular component. Further mechanisms involved are systemic coagulopathy and fibrin deposits, which form a matrix for the development of fibro-collagenous tissue. Besides the present work focusing on cellular plasma responses, previous studies showed that direct plasma treatment has significant impact on non-cellular components, such as blood hemostasis resulting in shortened full clotting time [42].”

Reviewer #2, Comment 2: Fibroblasts involved in peritoneal adhesions show increased proliferation and protein secration as well as reduced apoptosis rates.  It would be interesting to check how much plasma normalizes the excessive proliferation of fibroblasts activated, for example, by cytokines, and not calm state fibrablasts.

Authors’ response, Comment 2: We disagree with this comment:

  1. The most favorable concept of intraoperative PAM treatment is to prevent “calm-state” fibroblasts from getting activated and reprogrammed into adhesion-fibroblasts rather than treating already activated fibroblasts.
  2. We agree that investigating the PAM effects on activated fibroblasts (also referred to as myofibroblasts) is interesting. However, previous studies showed that the “activation status” of fibroblasts is not completely reliable and independent from cytokine activation in 2D cell culture systems (such as we used in this study) [x1]. The contact with surface polymers of Petri dishes alone may be followed by majoritarian activation of fibroblasts [x1].
  3. Therefore, experiments with reproducible and reliable fibroblast activation by cytokines should be performed in advanced “in-vivo-like” cell culture models like microfluidic organ-on-chip systems. We are in preparation for such experiments, but it is not context of the present work.

Due to the fact that Reviewer #2's question is reasonable, we would like to add his point to the discussion by means of the following paragraph:

“Moreover, the present study is limited to 2D cell culture experiments. Previous studies found the “activation status” of fibroblasts is not completely reliable in these systems. The contact with surface polymers of cell culture materials may be followed by majoritarian activation of fibroblasts [x1]. Due to this the 2D cell culture in-vitro data will have to be validated under in-vivo(-like) conditions.”

[x1]        Smithmyer ME, Cassel SE, Kloxin AM. Bridging 2D and 3D culture: probing impact of extracellular environment on fibroblast activation in layered hydrogels. AIChE J. 2019 Dec;65(12):e16837. doi: 10.1002/aic.16837. Epub 2019 Oct 16. PMID: 32921797; PMCID: PMC7480898.

Reviewer #2, Comment 3: There are also some small remarks

 «Forward- and side-scatter (FSC-H and SSC-H) characteristics were used to exclude dead cells» (Line 122-123). Dead cells can be eliminated using special dyes such as propidium iodide. In this case, it is debris - the remains of cells

Figure 1D it is not clear where the marker is. Please indicate either in the [the sentence ended here].

Authors’ response, Comment 3: We agree with Reviewer #2 and thank for his comment. We changed the sentence into:

“Forward- and side-scatter (FSC-H and SSC-H) characteristics were used to exclude debris.”

The marker in Figure 1D got further indicated.

Round 2

Reviewer 2 Report

The response of the authors to the remark is satisfactory.